# Specifying Goals to Deep Neural Networks with Answer Set Programming

**Primary Keywords:** *Learning*

## Abstract

Recently, methods such as DeepCubeA have used deep reinforcement learning to learn domain-specific heuristic functions in a largely domain-independent fashion. However, such methods either assume a predetermined goal or assume that goals will be given as fully-specified states. Therefore, specifying a set of goal states is not possible for learned heuristic functions while, on the other hand, the Planning Domain Definition Language (PDDL) allows for the specification of goal states using ground atoms in first-order logic. To address this issue, we introduce a method of training a heuristic function that estimates the distance between a given state and a set of goal states represented as a set of ground atoms in first-order logic. Furthermore, to allow for more expressive goal specification, we introduce techniques for specifying goals as answer set programs and using answer set solvers to discover sets of ground atoms that meet the specified goals. In our experiments with the Rubik's cube, sliding tile puzzles, and Sokoban, we show that we can specify and reach different goals without any need to re-train the heuristic function.

## Introduction

Deep reinforcement learning algorithms (Sutton and Barto 2018), such as DeepCubeA (McAleer et al. 2019; Agostinelli et al. 2019) and Retro* (Chen et al. 2020), have successfully trained deep neural networks (DNNs) (Schmidhuber 2015) to be informative heuristic functions. Combined with search methods such as A* search (Hart, Nilsson, and Raphael 1968), Q* search (Agostinelli et al. 2021), or Monte Carlo Tree Search (Kocsis and Szepesvári 2006), these learned heuristic functions can solve puzzles, perform retrosynthesis, as well as for compile quantum algorithms (Zhang et al. 2020). However, these DNNs do not generalize across goals where, in this context, a goal is a set of states in the state space that are considered goal states. Instead, these DNNs are either trained for a pre-determined goal or use methods such as hindsight experience replay (Andrychowicz et al. 2017) to generalize across pairs of start and goal states. As a result, specifying a goal to a DNN requires either training a DNN for that specific goal or obtaining the heuristic values for some representative set of goal states and taking the minimum heuristic value. This computationally burdensome process significantly reduces the practicality of DNNs for solving planning problems with dynamic goals. Furthermore, if one can only describe properties that a goal state

should or should not have, but does not know what states actually meet this criteria, obtaining a representative set of goal states is not possible.

To train DNNs to estimate the distance between a state and a set of goal states, we introduce DeepCubeA$_g$, a deep reinforcement learning and search method that builds on DeepCubeA (McAleer et al. 2019; Agostinelli et al. 2019) and hindsight experience replay (Andrychowicz et al. 2017) to learn heuristic functions that generalize across states and sets of goal states. Training data in the form of pairs of states and goals is obtained by starting from a given start state and taking a random walk to obtain a goal state. Given a process to convert a state to a set of ground atoms that represents what holds true in that state, we convert the obtained goal state to a set of ground atoms and then obtain a set of goal states by taking a subset of the set of ground atoms. We then train a heuristic function with deep approximate value iteration (DAVI) (Bertsekas and Tsitsiklis 1996; Agostinelli et al. 2019) to map states and sets of goal states to an estimated cost-to-go. When solving problem instances, we use the trained heuristic function with a batched version of A* search. We evaluate this approach on the Rubik's cube, 15-puzzle, 24-puzzle, and Sokoban (Dor and Zwick 1999) and results show that DeepCubeA$_g$ is able to find solutions for the vast majority of test instances and does so better than the domain-independent fast downward planner (Helmert 2006).

To allow for expressive goal specification, we build on the fact that goals are represented as sets of ground atoms. Therefore, to specify a goal, any specification language that can be translated to a set of ground atoms can be used. We choose answer set programming (ASP) (Brewka, Eiter, and Truszczyński 2011), a form of first-order logic programming, as the specification language because one can obtain stable models (Gelfond and Lifschitz 1988), also known as answer sets, for a given specification (answer set program) where each stable model is a set of ground atoms. Results show that diverse goals can be specified with simple answer set programs and reached using the learned heuristic function and search. An overview of our approach is described in Figure 1.

## Preliminaries

Our method builds on the DeepCubeA algorithm (Agostinelli et al. 2019) that was used to train a DNN as a heuristic function using deep approximate value iteration (Puterman and Shin 1978; Bertsekas and Tsitsiklis 1996). This heuristic function was then used in a batched version of weighted A* search (Pohl 1970) to solve puzzles such as the Rubik's cube and Sokoban. For specifying goals, we use ASP. In this section, we will describe the background of deep approximate value iteration as well as the background of ASP. We also describe the basics of the Rubik's cube.

### Deep Approximate Value Iteration

In the context of deterministic, finite-horizon, shortest path problems, approximate value iteration is a reinforcement learning (Sutton and Barto 2018) algorithm to learn a function, $h$, that maps a state $s$ to the estimated cost-to-go. The optimal heuristic function, $h^*$, returns the cost of a shortest path. The value iteration algorithm (Puterman and Shin 1978) takes a given $h$ and updates it to $h'$ according to Equation 1

$$h'(s) = \min_a(g^a(s, s') + h(s')) \tag{1}$$

where $g^a(s, s')$ is the cost to transition from $s$ to state $s'$ using action $a$ and $s'$ is the state resulting from taking action $a$ in state $s$.

In the tabular setting, value iteration has been shown to converge to $h^*$. However, for domains with large state spaces, such as the Rubik's cube, we do not have enough memory, or time, to do tabular value iteration. Therefore, we represent $h$ with a parameterized function $h_\phi$ with parameters $\phi$. The parameters of the function are trained to minimize the loss function in Equation 2

$$L(\phi) = (\min_a g^a(s, s') + h_{\phi^-}(s') - h_\phi(s))^2 \tag{2}$$

where $\phi^-$ are parameters of a target function that remains fixed for a certain number of training iterations and is periodically updated to $\phi$. This has been shown to make the training process more stable because the target remains stationary for extended periods of time (Mnih et al. 2015). When $h_\phi$ is a deep neural network, this approach is referred to as deep approximate value iteration (DAVI).

### Answer Set Programming

Answer set programming (ASP) (Brewka, Eiter, and Truszczyński 2011) is a form of logic programming that is built on the stable model semantics (Gelfond and Lifschitz 1988) which describes when a set of ground atoms, $M$, is a stable model, also known as an answer set, of a program, $\Pi$. Program $\Pi$ is restricted to be a set of rules in first-order logic of the form:

$$A \leftarrow B_1, ..., B_m, \neg C_1, ..., \neg C_n \tag{3}$$

where $A$, $B_i$, and $C_i$ are atoms in first-order logic. $A$ is in the "head", or the consequent, and $B_i$ and $C_i$ are in the "body",

or the antecedent. In this notation, $\neg$ represents negation, a comma represents conjunction, and $\leftarrow$ represents implication. Since all literals in the body are connected with conjunction, the body is true if and only if all literals in the body are true. Since the head has just has one atom, the head is true if and only if $A$ is true. Since the head and the body are connected by implication, the entire logical sentence is true if and only if one of the two following conditions are met: 1) the body is false; 2) the body is true and the head is true. If there are no literals in the body (also known as "facts"), then semantics dictate that the body is always true; therefore, the head must also always be true. If there are no atoms in the head (also known as "headless" rules), then semantics dictate that the head is always false; therefore, the body must also always be false. In practice, headless rules are used as constraints and are implicitly represented with a literal, $A$, in the head and a literal, $\neg A$, in the body that is in conjunction with the rest of the body literals. Therefore, headless rules are actually rules with negation in the body.

To determine if $M$ is a stable model of $\Pi$, we first must consider the grounded program of $\Pi$, which we will denote $\Pi_g$. To obtain $\Pi_g$, for all rules, $R$, in $\Pi$, every possible grounded version of $R$, $R_g$, is obtained and added to $\Pi_g$. A ground rule, $R_g$, is obtained from a rule, $R$, by substituting all variables in $R$ for a ground term appearing in $\Pi$. If there are no rules in $\Pi_g$ with negation, then there is one unique minimal stable model of $\Pi_g$ (Van Emden and Kowalski 1976; Gelfond and Lifschitz 1988) which corresponds to all atoms that are derivable from $\Pi_g$. An atom is derivable if it is in the head of a rule with a body that is true. If there are rules with negation in $\Pi_g$, then we can check if a given set of ground atoms, $M$, is a stable model of $\Pi_g$ by first computing the reduct (Marek and Truszczyński 1999) of $\Pi_g$ with respect to $M$, which we will denote $\Pi_g^M$. $\Pi_g^M$ is obtained by starting with $\Pi_g$ and deleting all rules that have a negative literal, $\neg C_i$, in the body if $C_i$ is in $M$ and then deleting all negative literals in the body of the remaining rules. $\Pi_g^M$ is now a negation free program, which means that it has one unique minimal stable model. If this stable model of $\Pi_g^M$ is equivalent to $M$, then $M$ is a stable model of $\Pi$. It should be noted that $\Pi$ can have multiple stable models if it contains negation. Furthermore, some ASP solvers, such as clingo (Gebser et al. 2022, 2014), allow for the use of disjunction, which can result in more than one stable model, even if negation is not present.

In ASP, choice rules may also be employed. Choice rules have a conjunction of literals in the body and a set of ground atoms in the head. If the body is true, then zero or more ground atoms in the head may be added to the stable model. For example for the following choice rule, if the body is true, then no ground atoms in the head may be added, one of the ground atoms in the head may be added, or both of the ground atoms in the head may be added (':-' indicates implication):

```
{a1(c,d); a2(d,c)} :- B_1, B_2, B_3
```

### The Rubik's Cube

The Rubik's cube is a three dimensional cube where each face of the cube consists of a 3 x 3 grid of stickers, which 54 stickers in total. Each sticker can be one of six colors: white, yellow, orange, red, blue, or green. These stickers combine where the faces intersect to form cubelets, where center cubelets have 1 sticker, edge cubelets have 2 stickers, and corner cubelets have 3 stickers. There are 6 center cubelets, 12 edge cubelets, and 8 corner cubelets. While the canonical goal state for the Rubik's cube is one where all stickers on each face have the same color, there are many other patterns that interest the Rubik's cube community (Ferenc 2013).

## Methods

### Learning Heuristic Functions for Goals

To learn a function that estimates the distance between a state, $s$, and a goal, $\mathcal{G}$, we must explicitly add the specified goal as an input to the heuristic function. Therefore, the heuristic function now becomes $h(s, \mathcal{G})$, that represents the cost to go from $s$ to a closest state in $\mathcal{G}$ when taking action $a$. We assume a function $G(s)$ that converts states to a set of ground atoms and some process to convert $\mathcal{G}$ to a representation suitable for the DNN. To train the DNN, we must first have the ability to sample state and goal pairs. From these pairs, we can then train the DNN using DAVI.

To sample state and goal pairs, the agent starts at a randomly generated state, $s_0$. The agent then takes $t$ actions, where $t$ is drawn from a random uniform distribution between 0 and a given number $T$. Each action is sampled according to a random uniform distribution[1]. The last observed state, $s_t$, is then selected to create a goal, $\mathcal{G}$, by first obtaining $G(s_t)$. Since any $G(s_t)$ that is a superset of a goal, $\mathcal{G}$, also represents a goal, we can simply randomly remove atoms from $G(s_t)$ to create $\mathcal{G}$ such that $\mathcal{G} \subseteq G(s_t)$ and; therefore, $s_t$ is a member of the set of goal states. The loss for the DNN is computed according to Equation 4. The parameters of the target network, $\phi^-$, are periodically updated to $\phi$. This training procedure is outline in Figure 1.

$$L(\phi) = (\min_a g^a(s, s') + h_{\phi^-}(s', \mathcal{G}) - h_\phi(s, \mathcal{G}))^2 \quad (4)$$

### Specifying Goals with Answer Set Programming

A logic program $\Pi$ used to specify a goal contains background knowledge, $B$, which is a set of rules that describes relevant domain knowledge, a goal specification, $H$, which is a set of rules with the atom `goal` in the head, a headless rule, `:- not goal`, that ensures `goal` must be true in all stable models, and a choice rule with an empty body that contains the set of all possible ground atoms, $K$, that can be used to represent a set of states. Given a stable model $M$ of $\Pi$, the subset of $M$ in $K$, $M_K$, represents a set of states. When obtaining a stable model, we would like to find a minimal $M_K$ to ensure the stable model is as general as possible. By minimal, we mean that removing atoms from

---

[1]Future work could use intrinsic motivation (Barto et al. 2004) to encourage the exploration of diverse states.

---

$M_K$ will result in `goal` no longer being true. To accomplish this, for an $M_K$ obtained from an answer set solver, we pick a ground atom, $a$, in $M_K$ and remove it. We check if `goal` can still be true for $M_K \setminus \{a\}$. If so, we set $M_K$ to $M_K \setminus \{a\}$ and repeat this process. If not, we choose another atom to remove. If we cannot remove any atoms and ensure `goal` is also true, then we terminate.

We will now formally define what a goal state and goal model is and how this relates to negation as failure.

**Definition 0.1** (Goal state). Given a program $\Pi$, a state, $s$, is a goal state if and only if $G(s)$ is a subset of some stable model of $\Pi$.

**Definition 0.2** (Goal model). Given a program $\Pi$, a set of ground atoms, $M$, is a goal model if and only if $M$ is a stable model of $\Pi$ and for every state, $s$, such that $G(s)$ is a superset of $M_K$, $s$ is a goal state.

If $M$ is indeed a goal model, then $M_K$ represents a set of goal states. However, it is not the case that all stable models of $\Pi$ are goal models since, in general, logic programs in ASP can exhibit non-monotonic behavior due to the closed world assumption. That is, a logic program is non-monotonic if some atoms that were previously derived can be retracted by adding new knowledge. To handle this, we will combine sampling and iteratively looking for larger models in an attempt to reduce the number of stable models that are not goal models. We will use the clingo (Gebser et al. 2014, 2022) ASP software package to specify goals.

### Reaching Goals

Given a DNN trained to estimate the distance between a state and a goal, where a goal is represented as a set of ground atoms, as well as a specification in the form of a logic program, $\Pi$, we can now describe how goals are reached. We first start by finding a stable model $M$ of $\Pi$. Since $M$ is not guaranteed to be a goal model, it is possible that the terminal state along some path to $M$ is not a goal state. Therefore, we will use the DNN with A* search to find one or more paths to $M$. If we find a terminal state that is a goal state, then we can return the path to that state. If we do not find any terminal state, then the stable model may represent a set of unreachable states (see the Future Work Section) and we sample a new stable model. Otherwise, if no terminal states are goal states, we will refine $M$ by searching for a stable model that contains a strict superset of $M_K$. This corresponds to finding a new stable model, $M'$, where $M'_K$ represents a subset of the states represented by $M_K$. To accomplish this, each atom in $M_K$ is added to $\Pi$ as a fact and a new stable model, $M'$, is found with the constraint that the size of $M'_K$ must be bigger than $M_K$. This process is outlined in Algorithm 1.

Similar to previous work (Agostinelli et al. 2019, 2021), to take advantage of the parallelism of graphics processing units (GPUs), we do a batched version of A* search that removes multiple nodes from the priority queue at each iteration.

Algorithm 1: Reaching a Specified Goal

**Input:** Program $\Pi$, DNN $h_\phi$, start state $s_0$, number of iterations $N$
**for** $i$ in $range(0, N)$ **do**
    Sample stable model $M$ of $\Pi$
    **while** $M$ is not None **do**
        $s_g$ = A*Search($s_0, M, h_\phi$)
        **if** $s_g$ is not None and $G(s_g)$ is a subset of some stable model of $\Pi$ **then**
            **return** $s_g$
        **else if** $s_g$ is None **then**
            Sample a new stable model $M$ of $\Pi$
        **else**
            Find $M'$ such that $M'$ is a stable model of $\Pi$ and $M_K \subset M'_K$
            $M = M'$
        **end if**
    **end while**
**end for**
**return** failure

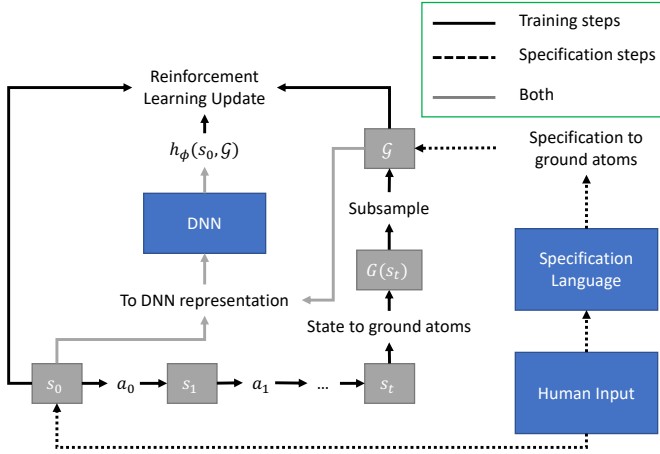

Figure 1: The figure outlines our training and goal specification procedure.

## Experiments

### Representation and Training

To specify a set of states for the Rubik's cube, we use a predicate, at_idx, of arity 2, that holds when a given color is at a given index. For example, at_idx(red,12) holds if a red sticker is at index 12 on the Rubik's cube. To represent a set of these ground atoms to the DNN, we first use a vector of length 54 to represent each sticker. We then set colors values of 0 through 5 based on the at_idx predicate. Unspecified indices in the vector are set to 6. We then use a one-hot representation of this vector as the input to the DNN.

To specify a set of states for the sliding tile puzzles, we use a predicate at_idx, of arity 3, that holds when a given tile is at a given x and y coordinate. To representation given to the DNN is a one-hot vector of tiles where there is a spe-

cial tile for those whose position is unspecified. To specify a set of states for Sokoban, we use the predicates agent, box, and wall of arity 3, which holds true if a given agent, box, or wall is at a given x and y coordinate. The representation given to the DNN is three binary matrices that represent the locations of the specified agent, boxes, and walls.

The architecture of the DNN and the optimization procedure is the same as that described in Agostinelli et al. (2019), with the exception that the parameters of the target network are updated based on a test set instead of a training loss. Specifically, we generate a test set on which we periodically test the greedy policy and an update is done when the number of states solved by the greedy policy increases. To randomly generate start states for the Rubik's cube, for each state, we start from the canonical goal state and randomly take between 100 and 200 actions. To generate start states for the sliding tile puzzle, we create random permutations and check for validity with parity. To generate start states for Sokoban, we start from a provided 900,000 training states (DeepMind 2018) and take a random walk with a length between 0 and 30. We set the maximum number of actions to take from the start state to generate goal states, $T$, to 30, for the Rubik's cube and 1,000 for the 15-puzzle, 24-puzzle, and Sokoban. We train and test using two NVIDIA Tesla V100 GPUs and 48 2.4 GHz Intel Xeon Platinum CPUs. Training is done with a batch size of 10,000 for 2 million iterations the Rubik's cube and 15-puzzle, four million iterations for the 24-puzzle, and one million iterations for Sokoban.

### Specifying Goals with Sets of Ground Atoms

To test the ability of DeepCubeA$_g$ to reach specified goals, we use the test states from Agostinelli et al. (2019), which contains 1,000 randomly generated states for the Rubik's cube, 500 randomly generated states for the 15-puzzle and 24-puzzle, and 1,000 randomly generated states for Sokoban. We use pattern databases (Culberson and Schaeffer 1998) to validate the cost of a shortest path. For the Rubik's cube, we use a pattern database that takes advantage of domain-specific mathematical group properties of the Rubik's cube (Rokicki 2016, 2010). We are using the 12 atomic actions for the Rubik's cube, so the maximum cost-to-go is 26. For the sliding tile puzzles, we use additive pattern database heuristics described in (Felner, Korf, and Hanan 2004).

For the test states for Rubik's cube and sliding tile puzzles, the goal is always the same. Therefore, we randomly generate 500 pairs of start and goal states by generating start states and taking a random walk with 1,000 to 10,000 steps, and using a random subset of the ground atoms obtained from the final state in the random walk to represent the goal. We compare DeepCubeA$_g$ to DeepCubeA, and the fast downward planning system (Helmert 2006) with the goal count heuristic, fast forward heuristic, and the causal graph heuristic. The PDDL domain files used can be found in the Supplementary Material.

For all test examples, we give each solver 200 seconds to solve them. DeepCubeA$_g$ is implemented in Python and uses two NVIDIA Tesla V100 GPUs for computing the heuristic function and a single 2.4 GHz Intel Xeon Platinum

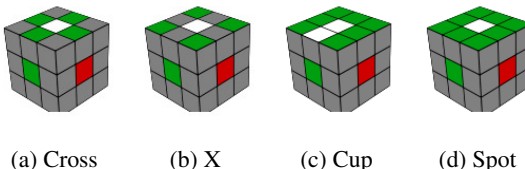

(a) Cross      (b) X      (c) Cup      (d) Spot

Figure 2: Examples of patterns that are combined to create goals.

CPUs, otherwise. Batch A* search is performed with a batch size of 10,000 for the Rubik's cube, 1,000 for the sliding tile puzzles, and 100 for Sokoban. Results are shown in Table 1. The results show that DeepCubeA$_g$ consistently outperforms the fast downward planning system in terms of the percentage of states that are solved. DeepCubeA$_g$ solves either 100% of states or close to 100% of states. In the single domain where the fast downward planner solved 100% of test cases, Sokoban, DeepCubeA$_g$ also solved 100% of test cases while also finding shorter paths. In cases such as the Rubik's cube and 24-puzzle for the canonical goal states, DeepCubeA$_g$ solves 100% of test states while the fast downward planner solves between 0% and 1.1%.

## Specifying Goals with Answer Set Programming

**Rubik's Cube**  We define colors, cubelets, and what color stickers the cubelets have. We also define directions (clockwise, counterclockwise, and opposite), faces, their colors (the same as the center cubelet), and their relation to one another (for example, the blue face is a clockwise turn away from the orange face with respect to the white face). We also describe what it means for a cubelet to have a sticker on a face as well as for a cubelet to be "in place" (all colors matching the center cubelet). We add constraints to the program to prune stable models that represent impossible states. These constraints include saying that different stickers from the same cubelet cannot be on the same face or opposite faces as well as saying that a cubelet cannot have a sticker on more than one face. The complete answer set program defining this is shown in the Supplementary Material.

To test our method, we draw from Ferenc (2013) to come up with goals that combine different Rubik's cube patterns shown in Figure 2. We also test our method with the canonical solved state for the Rubik's cube where all faces have a uniform color. Given the background knowledge, many patterns only require a few lines of code, as shown in the Supplementary Material. Note that the training procedure is not told of these patterns and is not aware that these patterns will be used for testing.

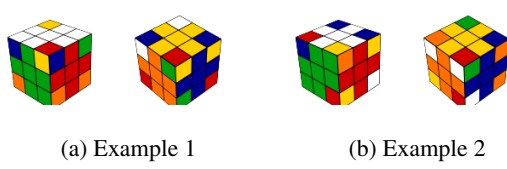

(a) Example 1          (b) Example 2

Figure 3: Reached goal of having a cross on all 6 faces where the center cubelet and cross are the same color.

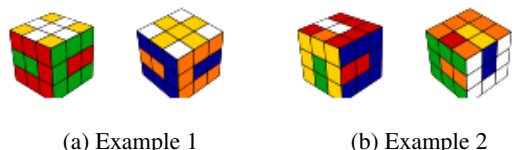

(a) Example 1          (b) Example 2

Figure 4: Reached goal of having cups on red, green, blue, and orange faces.

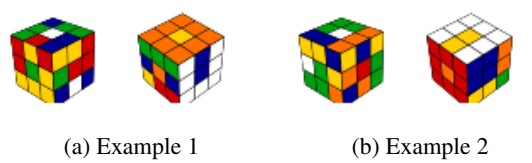

(a) Example 1          (b) Example 2

Figure 5: Reached goal of having a cup adjacent to a spot.

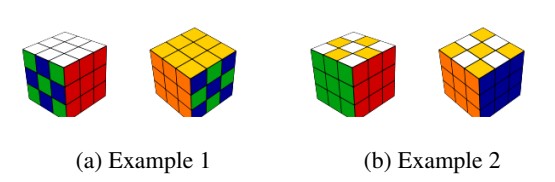

(a) Example 1          (b) Example 2

Figure 6: Reached goal of having two checkerboards on opposite faces with all of the other faces the same.

In addition to the canonical goal, we specify four other goals: (1) all faces have a cross where the cross is the same color as the center piece; (2) the red, green, blue, and orange faces have a cup on them (3) there is a spot adjacent to a cup with the opening of the cup facing the spot; (4) there are two checkerboard patterns (a cross combined with an X) on opposite faces and all other faces have uniform color.

Given a logic program, we use clingo to find stable models. We sample 100 randomly generated start states and use batch weighted A* search to find a path from these start states to a goal state. We use a batch size of 10,000 and a weight of 0.6 when doing batch weighted A* search. Each randomly generated start state is a given a budget of 50 iterations. If a goal is not found in that time, then a new stable model is generated. Visualizations of reached goals for the four non-canonical goals are shown in Figures 3, 4, 5, and 6. A table summarizing the time it takes to find stable models, solve all 100 start states, as well as the average path cost is shown in Table 2.

**Sokoban**  We also test our method on the Sokoban domain. This domain presents a unique challenge because the start state determines the ground atoms that will be present in a goal state. In particular, in the Sokoban domain, the walls cannot be modified; therefore, the specification of a goal must also take this into account. To address this, we add the location of the walls to the specification. The given background knowledge includes the dimensions of the grid, the relations of coordinates in terms of up, down, left, and right, what it means for a box to be immovable, what it means for a box to be at the edge of the grid, as well as

| Puzzle | Solver | Len | % Solved | % Opt | Nodes | Secs | Nodes/Sec |
|---|---|---|---|---|---|---|---|
| RC (canon) | PDBs⁺ | **20.67** | **100.00%** | **100.0%** | **2.05E+06** | **2.20** | **1.79E+06** |
| | DeepCubeA | 21.50 | **100.00%** | 60.3% | 6.62E+06 | 24.22 | 2.90E+05 |
| | DeepCubeA$_g$ | 23.00 | **100.00%** | 17.80% | 2.76E+06 | 51.08 | 5.40E+04 |
| | FastDown (GC) | - | 0.00% | 0.0% | - | - | - |
| | FastDown (FF) | **-** | 0.00% | 0.0% | - | - | **-** |
| | FastDown (CG) | **-** | 0.00% | 0.0% | **-** | **-** | - |
| RC (rand) | DeepCubeA$_g$ | 15.44 | **97.60%** | - | 1.92E+06 | 34.32 | 5.07E+04 |
| | FastDown (GC) | 7.18 | 32.80% | - | 2.67E+06 | 13.79 | 1.41E+05 |
| | FastDown (FF) | 6.49 | 31.20% | **-** | 4.87E+05 | 13.83 | 2.93E+04 |
| | FastDown (CG) | 7.85 | 33.80% | **-** | 1.12E+06 | 11.62 | 5.81E+04 |
| 15-P (canon) | PDBs | **52.02** | **100.00%** | **100.0%** | **3.22E+04** | **0.002** | **1.45E+07** |
| | DeepCubeA | 52.03 | **100.00%** | 99.4% | 3.85E+06 | 10.28 | 3.93E+05 |
| | DeepCubeA$_g$ | **52.02** | **100.00%** | **100.0%** | 1.81E+05 | 2.65 | 6.81E+04 |
| | FastDown (GC) | 36.75 | 0.80% | 0.80% | 9.05E+07 | 102.11 | 8.66E+05 |
| | FastDown (FF) | 52.75 | 80.80% | 24.80% | 2.92E+06 | 42.11 | 6.93E+04 |
| | FastDown (CG) | 41.95 | 4.40% | 1.20% | 2.00E+07 | 80.58 | 2.47E+05 |
| 15-P (rand) | DeepCubeA$_g$ | **33.98** | **100.00%** | - | **1.11E+05** | **2.39** | **6.07E+04** |
| | FastDown (GC) | 14.92 | 38.00% | - | 1.61E+07 | 18.77 | 5.46E+05 |
| | FastDown (FF) | 32.66 | 89.20% | - | 1.24E+06 | 17.39 | 5.65E+04 |
| | FastDown (CG) | 20.45 | 51.20% | - | 3.90E+06 | 21.41 | 1.20E+05 |
| 24-P (canon) | PDBs | **89.41** | **100.00%** | **100.00%** | 8.19E+10 | 4239.54 | **1.91E+07** |
| | DeepCubeA | 89.49 | **100.00%** | 96.98% | 6.44E+06 | 19.33 | 3.34E+05 |
| | DeepCubeA$_g$ | 90.52 | **100.00%** | 55.44% | **3.38E+05** | **6.08** | 6.36E+04 |
| | FastDown (GC) | - | 0.00% | 0.00% | - | - | - |
| | FastDown (FF) | 81.00 | 1.01% | 0.40% | 2.68E+06 | 89.84 | 2.91E+04 |
| | FastDown (CG) | - | 0.00% | 0.00% | - | - | - |
| 24-P (rand) | DeepCubeA$_g$ | 66.29 | **99.40%** | - | 2.49E+05 | 8.55 | 5.85E+04 |
| | FastDown (GC) | 9.86 | 10.00% | - | 9.54E+06 | 11.88 | 4.27E+05 |
| | FastDown (FF) | 26.35 | 26.00% | - | 5.99E+05 | 19.57 | 2.41E+04 |
| | FastDown (CG) | 13.75 | 12.60% | - | 1.42E+06 | 14.42 | 6.85E+04 |
| Sokoban | DeepCubeA | 32.88 | **100.00%** | - | **5.01E+03** | 2.71 | 1.84E+03 |
| | DeepCubeA$_g$ | **32.06** | **100.00%** | - | 1.77E+04 | 0.67 | 2.60E+04 |
| | FastDown (GC) | 31.94 | 99.80% | - | 3.17E+06 | 5.93 | 5.85E+05 |
| | FastDown (FF) | 33.15 | **100.00%** | - | 2.92E+04 | **0.32** | **7.49E+04** |
| | FastDown (CG) | 33.12 | **100.00%** | - | 4.43E+04 | 0.51 | 7.25E+04 |

Table 1: Comparison of DeepCubeA$_g$ with optimal solvers based on pattern databases (PDBs) that exploit domain-specific information and the domain-independent fast downward planning system with the goal count (GC) heuristic, fast forward heuristic (FF), and causal graph (CG) heuristic. Comparisons are along the dimension of solution length, percentage of instances solved, percentage of optimal solutions, number of nodes generated, time taken to solve the problem (in seconds), and number of nodes generated per second. For the Rubik's cube and sliding tile puzzles, experiments are done on canonical goal states (canon) and randomly generated goals (rand). For testing DeepCubeA on Sokoban, we report numbers obtained from the DeepCubeA GitHub repository[2].

basic constraints that state that two objects cannot share the same location. The predicates `agent(X,Y)`, `box(X,Y)`, and `wall(X,Y)` hold if an agent, box, or wall is at coordinates (X,Y). We investigate the following goals: (1) all boxes are immovable; (2) all boxes form a larger box; (3) the four boxes occupy the four corners next to the agent. Visualizations of reached goals are shown in Figures 7, 8, and 9. The answer set program for Sokoban is shown in the Supplementary Material.

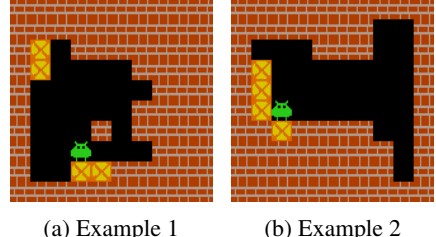

(a) Example 1          (b) Example 2

Figure 7: Reached goal where all boxes are immoveable.

Table 2: The time it takes to find stable models for each goal, the time it takes to find a path to the goal from 100 start states, and the average path cost.

|  | Stable Model Time (secs) | Solve Time (secs) | Path Cost |
|---|---|---|---|
| Canon | 0.33 | 625.62 | 23.82 |
| Cross6 | 0.35 | 218.45 | 11.50 |
| Cup4 | 11.17 | 1622.39 | 24.44 |
| CupSpot | 123.04 | 291.25 | 14.7 |
| Checkers | 0.44 | 602.03 | 24.00 |

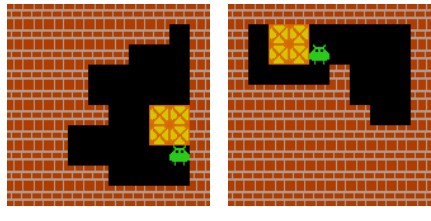

(a) Example 1        (b) Example 2

Figure 8: Reached goal where all boxes form a larger box.

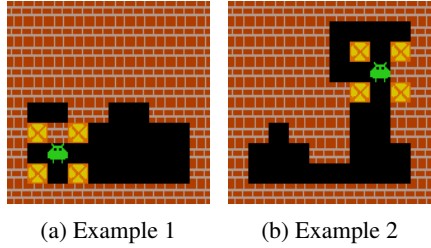

(a) Example 1        (b) Example 2

Figure 9: Reached goal where four boxes are at the four corners of the agent.

## Discussion

To illustrate the power of specifying a set of states as a goal instead of pre-determined states, we note that the Cross6 goal contains the canonical goal state in the set of states that it represents. However, finding the canonical goal state takes about three times as long and has a path cost that is about twice as long when compared to the Cross6 goal. This indicates that this method has the potential to allow us to discover more efficient plans as well as to discover new knowledge by achieving *unanticipated* goal states that even humans have not yet considered. For example, in a domain such as chemical synthesis, this could allow practitioners to discover new synthesis routes as well as learn more about chemistry by examining the properties of the unanticipated molecules that meet their specifications.

When examining solve time and path cost in Table 2, the Cross6 goal takes the least amount of time and has the shortest average path cost. The CupSpot goal is also comparable along these same metrics. This could be because Cross6 and CupSpot need to consider fewer stickers than other goals. However, the Cup4 goal takes the longest to reach out of all

the goals even though it also needs to consider fewer stickers than both the canonical goal and the Checkers goal. One indication of the cause of this is that A* search frequently went over its budget of 50 iterations for the Cup4 goal. This could be because some of the stable models actually represent sets of states that are not reachable. We discuss ways to overcome this in the Future Work Section.

When examining the time it takes to find stable models in Table 2, the CupSpot goal takes the longest out of all the goals. This could be due to having many constraints to consider when finding the stable models. However, dealing with the constraints when solving for the stable models could lead to faster solve times as fewer stable models will represent unreachable goals. It could also be the case that certain constraints could be expressed in a more concise manner.

## Related Work

Action Schema Networks (ASNets) (Toyer et al. 2020) are neural networks that exploit the structure of the PDDL language to learn a policy that generalizes across problem instances. However, ASNets are trained using imitation learning, which assumes a solver that can solve moderately difficult problems. On the other hand, we use reinforcement learning, which does not require that existence of any solver to learn. Furthermore, ASNets does not support arbitrary goal formulae. However, our approach of obtaining stable models from logic programs could be extended to descriptions of goals in PDDL and to ASNets. Furthermore, in the Future Work Section, we discuss ways goals can be represented with logic, itself, without having to solve for stable models.

Learning from partial interpretations (Fensel et al. 1995; De Raedt 1997) is a setting in inductive logic programming (Muggleton 1991; De Raedt 2008; Cropper and Dumančić 2022) where the training examples are not fully specified. This setting has also been applied to learning answer set programs from partial stable models (Law, Russo, and Broda 2014). This work has parallels with our work, except, instead of learning an answer set program as in Law, Russo, and Broda (2014), the specification is given in the form of an answer set program. Furthermore, instead of being given partial stable models as examples as in Law, Russo, and Broda (2014), the goal specification produces partial stable models that are then used by the DNN to reach the goal.

Research on training deep neural networks to generalize over both states and goals has mainly focused on goals that are represented by a single state. In reinforcement learning, Universal Value Function Approximators (Schaul et al. 2015) were proposed to learn a value function with an additional input of a goal state. Hindsight Experience Replay (Andrychowicz et al. 2017) built on this approach to learn from failures by using states observed during an episode as goal states, even if they were not the intended goal state. This approach has enabled learning in sparse reward environments, such as those involving object manipulation, and has shown to generalize to goal states not seen during training. After training, one can then specify what the goal state is, provided the practitioner has the ability to fully specify a goal state. However, this approach becomes impractical in

cases where there are a diverse set of acceptable goal states that the agent could possibly reach or where only high-level qualities of a goal are known, but the low-level details are not.

## Future Work

In this work, we investigated the Rubik's cube, which is a domain in which every state is reachable from every other state. However, in domains such as Sokoban, this is not the case. As a result, not all goals will be reachable from every possible start state. In these cases, the training process could be augmented by mining for "negative" goals (Tian et al. 2021) that cannot be reached. The DNN should then give a very high cost-to-go when a goal is not reachable from a given start state. We can then sample stable models that are below some threshold. This sampling procedure could also be imbued with a learned heuristic to guide the ASP solver towards reachable stable models.

In addition to unreachable goals, one could specify goals that only represent impossible states or have some stable models that only represent impossible states. While constraints could be manually added to the program to ensure no such stable models are found, preventing all such occurrences may require sophisticated domain-specific knowledge. Therefore, the AI system would ideally *discover* new constraints. Given a stable model that is thought to only represent impossible states, one could use inductive logic programming techniques and a generality relationship, such as entailment or theta subsumption (Plotkin 1972), to find the most general specification that only represents impossible states and add this to the background knowledge as a constraint.

Our approach of using ground atoms to represent a goal comes with the advantage of being agnostic to the specification language as long as it can produce a set of ground atoms. Therefore, in the case of using ASP as the specification language, changes can be made to the predicates or even the ASP software used without having to re-train the DNN. However, this comes with the computational cost of having to solve for a set of ground atoms given a specification. One could instead train the heuristic function to estimate the distance between a state and a lifted specification that either implicitly or explicitly contains variables. This could be done for any kind of specification, such as first-order logic or even natural language, given the ability to go from a state to a specification representing a set of states of which that state is a member. One could obtain training examples by obtaining a goal state and then searching for a first-order logic sentence that represents a set of states of which that goal state is a member. The downside to this approach is that any change in the vocabulary of the specification may require re-training of the DNN. Furthermore, this approach may but more representational burden on the DNN as it may need to implicitly consider stable models of a given specification.

## Conclusion

We have introduced DeepCubeA$_g$, a deep reinforcement learning and search method that trains DNNs to estimate the distance between a state and a set of goal states, where a set of goal states is represented as a set of ground atoms. Goals can be communicated to a DNN without the need to re-train the DNN for that particular goal and without the need for the DNN to see that particular goal during training. When compared to other domain-independent planners, DeepCubeA$_g$ consistently solved more test states and found shorter paths.

To allow for more expressive goal specification, we have formalized a method for specifying goals using a specification language that is accessible to humans. Furthermore, the language used to specify goals only needs to be able to be translated into a set of ground atoms, which makes the DNN agnostic to the specification language. Using answer set programming, one can easily specify properties that a goal state should or should not have without having to specify any goal state in particular. As a result, this method has the ability to discover novel goals and; therefore, facilitate the discovery of new knowledge.

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
