# OpenReview forum: "Specifying Goals to Deep Neural Networks with Answer Set Programming"
_icaps-conference.org/ICAPS/2024/Conference — ICAPS 2024_

### Official Review · Reviewer_hA5n · 2024-01-16

**Significance And Importance:** 2
**Soundness:** 3
**Novelty:** 3
**Clarity:** 3
**Overall Evaluation:** 1
**Confidence:** 3

**Weaknesses:**

2: No major or minor weaknesses.

**Contributions Of The Paper:**

The paper proposes a deep reinforcement learning + search method, DeepCubeAg.
The method learns heuristic functions that generalize over goals.
For solving instances, DeepCubeAg queries the learned heuristic with (batched) A*.
The algorithm builds on DeepCubeA (McAleer et al. 2019; Agostinelli et al. 2019), which does not generalize over goals.
In addition, the paper introduces techniques for (1) specifying goals as answer set program (ASP) and (2) compute sets of ground atoms to fulfill the program specification.
In an empirical evaluation, DeepCubeAg is shown to produce goal-generalizing heuristic functions that outperform FD baselines and that are competitive to DeepCuebA.

**Ethical Considerations:**

(1) Not Applicable: The paper does not have any ethical considerations to address

**Nomination For Best Paper:**

No

**Questions For Authors:**

(1) Can you think of alternatives to ASP (i.e., for the case where the heuristic still receives ground atoms, and is not trained directly on the specification)?

(2) You reuse the DNN architecture by Agostinelli et al. (2019).
Yet your networks generalize over goals, so I would expect there are at least some modification/adaptions.
Can you elaborate?

(3) I am somewhat struggling with your reasoning about "unanticipated goal states" (line 442 and following).
Concerning your example: Cross6 describes a set of a goal states that subsumes the canonical goal state.
How, does this qualify as "achieving unanticipated goal states"?
It does not seem all that surprising that for the former a shorter path can be found more quickly.
We obtain this result because we consider an extended set of goal states (relative to Canon).

(4) What is the motivation of/rational behind visualizing "reached goals" for Sokoban (Figure 7 through 9)?

(5) Will you publish your code? I can find some data files in the supplementary material. But I cannot find (or cannot recall to have found) any information concerning publication of code.

Concerning presentation: I feel like some examples on ASP might be helpful.

Possible Typos:

"h(s, G), that represents the cost to go from s to a closest state in G when taking action a" (l. 203) Shouldn't this be independent of the action?

"which [with] 54 stickers in total" (l. 190)

"To[The] representation given to the DNN" (l. 305)

"Furthermore, this approach may but [put] more representational burden on the DNN" (l. 570)

**Reproducibility:**

4: Authors promise to release code and domains (whichever apply).

**Strengths Of The Paper:**

The paper introduces a RL technique to learn heuristics that generalize over goals.
This novel technique is in line with the nature of the PDDL, i.e., given a problem domain, goal can be specified on a per-instance basis.

The ASP specification of goals improves expressiveness.
In addition, (on the upside, see below) the underlying deep NN heuristic is, in principle, agnostic to the specification language provided it can produce a set of ground atoms.

**Weaknesses Of The Paper:**

On the downside (see above), the learned policy being agnostic to the specification language adds another layer of computational cost to compute a set of ground atoms.
In addition, given the lack of a competitor, the choice of ASP as specification language remains -- while theoretically motivated -- empirically arbitrary.

---

> ### Author Rebuttal · Authors · 2024-01-26
>
> We thank the reviewer for their detailed feedback which we will incorporate in the final version of the paper.
>
> > Alternatives to ASP
>
> Yes, one could use Prolog to specify goals where goals are prolog terms. These terms could then be translated to ground atoms, but the translation process may need to be domain-dependent.
>
> > …agnostic to the specification language adds another layer of computational cost…
>
> We agree that this is one potential drawback of our work, and we discuss this in the Future Work section on line 555. We leave this to future work.
>
> > Reuses the DNN architecture by Agostinelli et al. (2019) but generalizes over goals.
>
> The architecture (layers, layer widths, depth, etc.) is the same. The only modification is an additional input for the goal. However, the values of the parameters are different because it is trained from scratch (random initialization) using techniques described in the Learning Heuristic Functions for Goals subsection.
>
> > "unanticipated goal states"
>
> We now realize this language is imprecise. To rephrase: We can specify a set of goal states and reach an element from that set that we did not know about previously. For example, for the Cross6 goal, we know the canonical goal state would be an element of the set; however, Figure 3 shows that we reached other elements in this set that we were not explicitly aware of, beforehand.
>
> > What is the motivation of/rational behind visualizing "reached goals" for Sokoban
>
> So that the reader can verify that the specified goals were, indeed, reached.
>
> > Will you publish your code?
>
> Yes, the code will be made freely and publicly available.
>
> > Concerning presentation: I feel like some examples on ASP might be helpful.
>
> We agree with the reviewer and have included the ASP files in the supplementary material. We hope these files can show the ease with which one can specify goals given background knowledge. For example, specifying Cross6 is only two lines:
>
> cross(F, CrossCol) :- face(F), color(CrossCol), #count{Cbl: edge_cbl(Cbl), onface(Cbl, CrossCol, F)} = 4.
>
> cross6 :- #count{F: cross(F, CCol), face_col(F, CCol)} = 6.
>
> > "h(s, G), that represents the cost to go from s to a closest state in G when taking action a" (l. 203) Shouldn't this be independent of the action?
>
> Yes, we will fix this in the updated version.
>
> > Other typos
>
> We will fix these in the updated version.

---

### Official Review · Reviewer_c9Ke · 2024-01-23

**Significance And Importance:** 2
**Soundness:** 3
**Novelty:** 3
**Clarity:** 3
**Overall Evaluation:** 1
**Confidence:** 3

**Weaknesses:**

1: Minor weaknesses that are easily fixable.

**Contributions Of The Paper:**

The paper introduces the use of goals to train better DNN heuristic functions for classical planning.  Goals are introduced in the form of ASP formulae, which guide the learning.  The approach is evaluated on several classical planning benchmark problems: rubik's cube, 15-tile, 24-tile, and sokoban.  The results show that the learned heuristic performs similar to the original DeepCubeA solver.  In the RC and tile problems (which have no dead ends) an optimal classical planner does not solve many problems.  In the Sokoban problems (which may have deadends) the planner and DeepCubeA approaches seem comparable.

-----
Thank you for the clarifications.

**Ethical Considerations:**

(1) Not Applicable: The paper does not have any ethical considerations to address

**Nomination For Best Paper:**

No

**Questions For Authors:**

I noticed that DCA was missing from the table for RC (rand), 15-P (rand), 24-P (rand).  Is that because these weren't solved by DCA?

**Reproducibility:**

5: Code and domains (whichever apply) are already publicly available

**Strengths Of The Paper:**

Overall, this is a very interesting idea.  The paper resolves many interesting technical challenges to get a working pipeline of using the goals to train the heuristic function.

**Weaknesses Of The Paper:**

The paper keeps making comparisons between Fast Downward and DeepCubeAg, but I don't actually think that's the key contribution worth noting for several reasons: First off, the comparison is marginally unfair because ASP isn't being used to "help" Fast Downward  (suppose you could, what might be the result?)  Second, the learned heuristic could be used within a planner to speed up the planner.  In other words, what if you used the heuristic within Fast Downward.  Finally, it's more worthwhile to compare DeepCubeA to ..Ag, since the change between these two systems is the main result.  In that case, the results are more mixed, *unless* the missing rows in the table mean that DeepCubeA could not solve the problems?

---

> ### Author Rebuttal · Authors · 2024-01-26
>
> We thank the reviewer for their detailed feedback which we will incorporate in the final version of the paper.
>
> > The paper keeps making comparisons between Fast Downward and DeepCubeAg…First off, the comparison is marginally unfair because ASP isn't being used to "help" Fast Downward
>
> The goals in Table 1 (both canon and rand) are specified with a set of ground atoms; therefore, ASP is not used for Table 1 as it is used for the section starting at line 380. Table 1 shows that, given a set of ground atoms (as opposed to finding one with ASP), DeepCubeAg compares favorably to existing solvers. We display the results of these experiments before the ones using ASP to show that, since ASP is used to find a set of ground atoms, DeepCubeAg is the best candidate for being used in combination with ASP. We will clarify this in the updated version.
>
> > Second, the learned heuristic could be used within a planner to speed up the planner. In other words, what if you used the heuristic within Fast Downward.
>
> We agree that this is possible and will mention this possibility in the future work section.
>
> > Finally, it's more worthwhile to compare DeepCubeA to ..Ag, since the change between these two systems is the main result. In that case, the results are more mixed, *unless* the missing rows in the table mean that DeepCubeA could not solve the problems?... I noticed that DCA was missing from the table for RC (rand), 15-P (rand), 24-P (rand). Is that because these weren't solved by DCA?
>
> The reviewer is correct, DeepCubeA is missing in the (rand) experiments because it is trained for a pre-determined goal (the canonical goal) and, therefore, cannot solve these problems. For DeepCubeA to solve a new goal, it would have to be re-trained, which could take hours to days. We will make this shortcoming of DeepCubeA more explicit in the revised version.

---

### Official Review · Reviewer_heRi · 2024-01-23

**Significance And Importance:** 1
**Soundness:** 3
**Novelty:** 2
**Clarity:** 4
**Confidence:** 4

**Weaknesses:**

1: Minor weaknesses that are easily fixable.

**Contributions Of The Paper:**

This paper extends the work of DeepCubeA for learning domain-specific heuristic for a fixed problem, with a predetermined goal, to propose a new method that is able to learn good heuristics for different goals (problems) without having to retrain.

**Ethical Considerations:**

(1) Not Applicable: The paper does not have any ethical considerations to address

**Nomination For Best Paper:**

No

**Overall Evaluation:**

-1: (weak reject)

**Questions For Authors:**

-  it is not clear the reason for using ASP to specify the goal. Why not using first order logic instead or even PDDL? It is not clear also what is the need for a sable model. One can define a stable model in FOL.
- the call for an ASP solver might be too costy.

**Reproducibility:**

4: Authors promise to release code and domains (whichever apply).

**Strengths Of The Paper:**

- The proposed method uses RL to learn heuristics for different planning problems, with different initial and goal states

**Weaknesses Of The Paper:**

- the contribution is incremental.
- the experimental validation was done over a classical planner. Although FastDoward is considered an efficient classical planning system,  the authors should compare with other works that also try to generalize the learned model for different tasks, such as ASNET.
- the generalization over different goal formulae were not validated. There is no experiments showing how the learned heuristic is efficient (or not) for goals not considered in the training phase.
- it is not clear the reason for using ASP to specify the goal.
- the use of 1 million of states for the sokoban game seems too high (pag 4)

---

> ### Author Rebuttal · Authors · 2024-01-26
>
> We thank the reviewer for their detailed feedback which we will incorporate in the final version of the paper.
>
> > Comparison to ASNets
>
> We discuss ASNets in the related work section. Of note, (1) ASNets assume the existence of a solver that can solve moderately difficult problems, while we do not because we use reinforcement learning to learn a heuristic function; (2) ASNets does not support arbitrary
> goal formulae while our approach does because it is agnostic to the specification language given a conversion process. However, we note that our approach of using ASP to find a set of ground atoms could be used to generate goals in PDDL and, thus, be applied to ASNets.
>
> > generalization over different goal formulae
>
> Table 1 and the Section starting at line 380 show results for how our approach generalizes over goals. We would like to re-emphasize that this data was generated independently of the training data; therefore, the training process was not aware of these goals.
>
> > it is not clear the reason for using ASP to specify the goal. Why not using first order logic instead or even PDDL? It is not clear also what is the need for a sable model. One can define a stable model in FOL.
>
> The reason we use ASP is because answer set solvers allow us to obtain a set of ground atoms and our approach also uses a set of ground atoms to specify goals. It is true that alternative languages, such as Prolog or PDDL, could be used to specify goals, which we highlight by emphasizing that our approach is agnostic to the specification language used given it can be converted to a set of ground atoms (Figure 1). We believe this to be an advantage of our approach.
>
> > the use of 1 million of states for the sokoban game seems too high
>
> We are not sure by what metric the reviewer deems this to be too high. We are simply using an existing dataset [1] which has been used to evaluate multiple solvers [2,3].
>
> > the call for an ASP solver might be too costy.
>
> We agree that this is one potential drawback of our work, and we discuss this in the Future Work section on line 555. We leave this to future work.
>
> [1] DeepMind. 2018. boxoban-levels. https://github.com/
> deepmind/boxoban-levels/tree/master/unfiltered
>
> [2] Racanière, Sébastien, et al. "Imagination-augmented agents for deep reinforcement learning." NeurIPS (2017).
>
> [3] Agostinelli, Forest, et al. "Solving the Rubik’s cube with deep reinforcement learning and search." Nature MI (2019).

---

### Meta-Review · Area_Chair_rQ44 · 2024-02-06

**Recommendation:** Accept (Poster)
**Confidence:** 3

**Metareview:**

The reviewers agree that the topic of the paper is interesting, useful, and fairly novel. Issues were raised about the motivations for using ASP as a language for goal specification, the experiments, and the choice of the planning tools. None of these appears critical alone, but taken all together, they contribute to weakening the contribution. Nonetheless, some of the concerns were addressed satisfactorily by the authors in their  response, and it is reasonable to expect that the responses will be implemented in the final version, in case of acceptance. Under this assumption, the work reaches the quality standards of an ICAPS paper.

**Ethical Considerations:**

(1) Not Applicable: The paper does not have any ethical considerations to address